# Extraction of Uranium in Nitric Media with Novel Asymmetric Tetra-Alkylcarbamide

**DOI:** 10.3390/molecules27175527

**Published:** 2022-08-27

**Authors:** Qi Chen, Baole Li, Junli Wang, Haowei Zhu, Xiwen Chen, Yifu Hu, Jia Zhou, Xiang Li, Weifang Zheng, Taihong Yan

**Affiliations:** China Institute of Atomic Energy, P.O. Box 275-26, Beijing 102413, China

**Keywords:** uranium, extraction, tetra-alkylcarbamides, DFT calculation

## Abstract

The use of tetra-alkylcarbamides as novel ligands: N,N-butyl-N’,N’-hexylurea (***L1*: ABHU**), and N,N-butyl-N’,N’-pentylurea (***L2*: ABPU**), for the solvent extraction and complexation behaviors of uranium(VI) was synthesized and investigated in this study. The effects of HNO_3_ and NO_3_^−^ concentrations in the aqueous phase on the distribution ratio of U(VI) were examined. Under 5 mol/L HNO_3_ concentration, *D_U_* reached 5.02 and 4.94 respectively without third-phase formation. During the extraction, slope measurements and IR spectral analysis revealed that the U(VI) complexes are a form of UO_2_(NO_3_)_2_·2***L*** for both ligands. In addition, thermodynamic studies showed that the uranium extraction reaction was a spontaneous exothermic reaction. The deep structural analysis of the complexes was realized with DFT calculation. The bond length, bond properties, and topology of the complexes were discussed in detail to analyze the extraction behavior. This study enriches the coordination chemistry of U(VI) by tetra-alkylcarbamides, which may offer new clues for the design and synthesis of novel ligands for the separation, enrichment, and recovery of uranium in the nuclear fuel cycle.

## 1. Introduction

The sustainable production of energy is one of the most important scientific and technological challenges facing humanity [1]. Although research into alternative energy sources has been prompted by environmental concerns and the growing global population, the majority of the world's energy until now has continued to be produced by burning fossil fuels. Nuclear energy plays an important role in the search for alternative energies to fossil fuels [2]. Theoretically, one gram of ^235^U may provide as much energy through nuclear fission as 1.5 million kilograms of coal burned. The development of innovative extraction materials to enhance the nuclear fuel cycle for uranium separation, enrichment, and recovery is greatly facilitated by nuclear power.

To date, various ligands such as tri-n-butyl phosphate (TBP) [3,4], trialkyl phosphine oxide (TRPO) [5], octyl-(phenyl) N, N-diisobutyl carbamoyl methyl phosphine oxide (CMPO) [6], tris-2-ethyl hexyl phosphate (TEHP) [7], Di(2-ethylhexyl) pivalamide (D2EHPVA) [8], etc. have been reported for the effective extraction of uranyl dication (UO_2_^2+^). However, P-containing ligands would seriously pollute the ecosystem, and the selective extraction capacity of U(VI) also needs to be improved.

Other ligand families are being investigated [9,10,11,12,13], among which N,N-dialkyl amides (or monoamides) are gradually showing the potential for U(VI) extraction [14,15]. On this basis, tetra-alkylcarbamides were developed. Compared to monoamides, tetra-alkylcarbamides have received little attention despite the first report by Siddall’s group in the 1960s [14]. In recent years, tetra-alkylcarbamides have returned to the public view, their composition of only C, H, O, and N, which can reduce the volume of secondary waste [16], their ability to extract uranium(VI) from nitric acid [17], their high selectivity for uranium(VI) versus plutonium(IV) and fission products, and the nature of their radiolysis and hydrolysis products make them very valuable for research.

Research has been conducted with four different alkyl chains lengths of tetra-alkylcarbamides by Berger et al [18,19] (TBU: N,N,N’,N’-tetra-n-butylurea, TPU: N,N,N’,N’-tetra-n-pentylurea, THU: N,N,N’,N’-tetra-n-hexylurea, TOU: N,N,N’,N’-tetra-n-octylurea). Tetra-alkylcarbamates with four straight chains had good actinide (IV) and (VI) extraction performance and a greater uranyl extraction rate than reference monoamides (MDEHA: N-methyl-N-decyl-(2-ethyl)hexanamide). Two ligands showed a very high U(VI) loading capacity without any third phase formation even at high HNO_3_ concentration, and their viscosity remained acceptable for process development. As a result, their potential for use in uranium (VI) extraction in the post-treatment process was demonstrated.

Theoretically, the design of four different substituent structures (combination of long and short carbon chains) can increase the solubility of the complex in the diluent while maintaining a high distribution ratio thereby reducing or preventing the generation of a third phase during extraction [20]. However, the studies on asymmetric tetra-alkylcarbamides are rather limited, only Vats et al. synthesized a ligand named N,N-diethyl-N',N'-diisobutylurea [21], which exhibits superior selectivity for uranium in the presence of interfering thorium and other lanthanide ions from nitric acid media.

Thus, to enrich the understanding of asymmetrical tetra-alkylcarbamides, the present work synthesized and studied the extraction and coordination behaviors of uranium with two novel ligands: N,N-butyl-N’,N’-hexylurea (***L1***: **ABHU**), and N,N-butyl-N’,N’-pentylurea (***L2***: **ABPU**)(structure describe in Figure 1). Batch studies in 0.5 mol/L ***L***/n-dodecane (***L*** stands for these two ligands in this study) were used to explore solvent extraction, and Fourier transform infrared spectroscopy (FT-IR) was used to investigate the complexation behaviors between ***L*** and U(VI). The extraction thermodynamic data were also determined. Furthermore, DFT calculations were performed to define uranium(VI) speciation in the organic phase and to better understand the extraction behavior of ***L*** for U(VI). This study would provide guidance for the further exploration of the novel U(VI) extraction ligands.

## 2. Experimental

### 2.1. Chemicals and Reagents

The China Institute of Atomic Energy (CIAE) provided all uranium elements for the experiments, which were utilized without additional purification. All of the radioactive experiments (with uranium) were carried out in a radiological facility that followed established safety protocols.

The diluent utilized was n-dodecane, which was bought from Macklin. Aladdin provided the nitric acid as well as all the reagents used in the ligand synthesis section. All of these reagents are more than 98% and ready to use without further purification.

### 2.2. Synthesis of the ligands

The synthesis procedure for the **ABHU** and **ABPU** is given in Figure 2. The purity of all the ligands was tested by a high-resolution mass spectrogram, nuclear magnetic resonance hydrogen spectrum (^1^H-NMR), and nuclear magnetic resonance carbon spectrum (^13^C-NMR) (purity > 99%) (more detail in Appendix A).

### 2.3. Solvent Extraction Procedure

Except for the studies addressing the dependency of the ligand concentration, the concentration of ligand in the organic phase was 0.5 mol/L. Except for the studies concerning the influence of HNO_3_ concentration, all of the experiments used a 5 mol/L HNO_3_ concentration. The initial concentration of U(VI) was 10 g/L.

The dependence of HNO_3_ concentration: 0.1 mol/L, 0.2 mol/L, 0.5 mol/L, 1 mol/L, 2 mol/L, 3 mol/L, 5 mol/L, 7 mol/L, 8 mol/L. The dependence of NaNO_3_ concentration: 0.1 mol/L, 0.2 mol/L, 0.5 mol/L, 1 mol/L, 2 mol/L, 3 mol/L.

Before the extraction tests, an organic phase containing 0.5 mol/L ligands in n-dodecane was pre-equilibrated twice with HNO_3_, which has the same volume (1 mL) and concentration as the aqueous phase in the next extraction experiment. Pre-equilibrated ***L***/n-dodecane and different concentrations of HNO_3_ and NaNO_3_ containing UO_2_^2+^ were utilized as the organic and aqueous phases, respectively. By combining equal amounts of organic and aqueous phases, solvent extraction was achieved (1 mL). For all of the extraction experiments, the temperature was held at 26 °C ± 1 °C in a constant-temperature vibrator for 30 minutes (extracted completely). A centrifuge was used to separate the organic and aqueous phases (5 minutes, 2000 rpm/min).

Two aliquots of the aqueous raffinate were collected for concentration determination during double extraction procedures under the same experimental conditions. As a result, each distribution ratio was calculated by averaging several samples, with an error of less than 5%.

The distribution ratios of U(VI) can be defined as:(1)$$D_{U}=\frac{C_{o}}{C_{a}}=\frac{V_{a}{(C}_{\mathrm{ini}}{-C}_{a})}{V_{o}C_{a}}=\frac{{(C}_{\mathrm{ini}}{-C}_{a})}{C_{a}}$$
where C_ini_, C_a_, and C_o_ represent the initial concentration of metal ions in the aqueous phase, in the raffinate, and in the organic phase, respectively. V_a_ and V_o_ denote the volumes of the aqueous and the organic phases, respectively.

### 2.4. Analytical Techniques

During the studies, the concentration of U(VI) was determined by ICP (Thermo Scientific model iCAP7400 Duo, Waltham, MA, USA). Each sample was repeated two times, with the average utilized for further study. In addition, all the uranium concentrations in the organic phase were calculated by subtracting the raffinate concentration from the initial metal concentration.

The coordination characteristics of ***L*** and the extracted U(VI) complexes were studied using Fourier transform infrared spectroscopy (FT-IR) using a Shimadzu Iraffinity-1s Fourier transform infrared spectrometer coupled with the OMNIC 5.1 software to collect and analyzes the spectra.

### 2.5. Computation Details

The geometry optimizations of all the species were computed at the density functional theory (DFT) level with the Gaussian 09 program using the hybrid B3LYP functional in both the gas and solution phases without symmetry constraints [22].

The ECP60MWB coupled with ECP60MWB-SEG valence basis sets were utilized for uranium, where 60 inner-shell core electrons were replaced by an effective core potential (ECP) created for a neutral atom and the remaining 32 electrons were represented by the corresponding valence basis set. The 6-311g(d,p) basis set was used for the other light atoms C, H, O, and N. The polarizable continuum model was used to assess the solvation effect (Hexane) (PCM).

On the optimized geometries, vibrational frequency calculations (without imaginary frequencies) were performed at the same level of theory to corroborate the identified stationary spots as minima on the potential energy hypersurface. Multiwfn software was also used to calculate Hirshfeld charges and Wiberg bond indices [23].

The topological analysis of electron density has been investigated utilizing quantum theory of atoms in molecules (QTAIM) and Multiwfn software to access the bonding characteristics of metal-ligand bonds. A chemical bond is characterized by the presence of a maximum electron density line along a bond path between each atom pair and the bond critical point in the QTAIM technique. Therefore, it can give useful information regarding the characteristics of chemical bonds, [24,25] which have been frequently applied to actinide complexes research [26].

## 3. Results and Discussion

### 3.1. Solvent Extraction

#### 3.1.1. Effect Nitric Acid Concentration

Figure 3a shows the *D_U_* in the ***L***/n-dodecane system as a function of HNO_3_ concentration. For both ligands, as the nitric acid concentration increases from 0.1 to 8 mol/L, the *D_U_* increases at first, then decreases. The peak value of *D_U_* appears at 5 mol/L nitric acid concentration and is 5.02 and 4.94 for ***L1*** and ***L2***, respectively. Even under high HNO_3_ concentration (>5 mol/L), no third phase is formed, and their viscosity remains acceptable for experiments.

For ***L1*** with a longer alkyl chain, as illustrated in Figure 3a, although slight, the *D_U_* becomes bigger and follows the sequence ***L1***>***L2***. For both monoamides [27,28,29] and tetra-alkylcarbamides, [18] the effect of alkyl chain length has been studied previously. For example, for the extraction study of N-methyl, N-butyl derivatives of hexanamide (MBHA), octanamide (MBOA) and decanamide (MBDA), the extraction of the U(VI) follows the order of MBHA < MBOA < MBDA. A longer carbon branch chain can improve the solubility of the complex in the organic phase, in order to boost the *D_U_*.

Figure 3b depicts the influence of NO_3_^−^ on the *D_U_*. The *D_U_* increases with the increase in ionic strength. In light of the findings in Figure 3b, it is anticipated that the salting-out effect and co-ionic action of NO_3_^−^ increased the complexation of ligands and U(VI), resulting in a rise in *D_U_*. As the concentration of HNO_3_ in the extraction system rises to 5 mol/L, H^+^ competition with U(VI) in the extraction process reduces the amount of free ligands molecules available for complexation with U(VI), resulting in a lower *D_U_*.

In contrast to the monoamide N-methyl-N-decyl-(2-ethyl)hexanamide(MDEHA) extraction study, 1.2 mol/L of MDEHA in TPH was observed with *D_U_* = 9 at 4 mol/L HNO_3_ concentration and *D_U_* = 0.4 at 0.5 mol/L HNO_3_ concentration. Compared with similar carbon chain length symmetric ligand N,N,N’,N’-tetra-n-butylurea (TBU), 1.2 mol/L TBU in the TPH diluent system with a nitric acid concentration of 0.5 mol/L had *D_U_* = 0.6. The results of this study are very competitive, especially since they were achieved at a ligand concentration of only 0.5 mol/L [18].

#### 3.1.2. Complexation of U(VI) with ***L***

U(VI) transfer from aqueous phase into organic phase involve co-extraction of NO_3_^−^ from this we propose a neutral extraction mechanism for both ligands as follows:(2)$${\mathrm{UO}}_{2}^{2+}{}_{a}{+2\mathrm{NO}}_{3}^{-}{}_{a}{+\mathrm{nL}}_{o}\to {\mathrm{UO}}_{2}{{(\mathrm{NO}}_{3})}_{2}\cdot {\mathrm{nL}}_{o}$$

In this equation, n is the number of two ligands molecules coordinated during the extraction. The subscripts a and o denote the aqueous and organic phases, respectively.

Typical slope analysis of the extraction equilibrium data was used to estimate the extracted stoichiometry of complexes [30]. The *D* values of U(VI) were shown as a function of ligand concentration at 5 mol/L nitric acids. Two ligands had slope values of 1.87 and 1.83, respectively, as shown in Figure 4, indicating that the stoichiometric ratio of U(VI) to ***L*** in the extracted species is mainly in the form of 1:2, as the major complexation formed as UO_2_(NO_3_)_2_·2***L***.

#### 3.1.3. Uranium Loading Capacity

The research on the maximum uranium loading capacity in organic phases was carried out by 0.5 mol/L ***L*** in organic solutions which were contacted four times with fresh uranium solution of U(VI) 43.9 g/L (0.18 mol/L) in 5 mol/L nitric acid. The uranium loading concentration after each cycle is reported in Figure 5. 

For the ***L1*** ligand with longer carbon chains, no third phase occurred in all four cycles, and the loading of U in the organic phase reached 50.3 g/L after four cycles. For the ***L2*** ligand with a shorter carbon chain, third phase appeared after the third cycle, and the loading of U in the organic phase after the second cycle was 44.3 g/L. Compared to the reported results, 1.2 mol/L TBU showed a third phase in the first cycle of contact with 200 g/L uranium [18]. The asymmetry of the ligands seems to have mitigated the third-phase generation.

As much as 0.21 mol/L (50.3 g/L) of uranium can be extracted in organic phases, which corresponds to a C***L1***/CU(VI) ratio of 2.4. This result indicates that the ***L1*** should form a 1:2 uranyl/carbamide complex as major species in the organic phase, which is consistent with the results obtained by slope analysis.

### 3.2. FTIR Studies

The IR spectrum investigations were carried out as presented in Figure 6. The main characteristic band of ***L*** is the C=O_free_ stretching band at 1645 cm^−1^. After contact with a 3 mol/L nitric acid aqueous phase, another ${C=O}_{{\mathrm{HNO}}_{3}}$ stretching band appeared at 1570 cm^−1^ for ***L1*** and at 1567 cm^−1^ for ***L2***, which is indicative of a more stable C=O bond. It may be caused by the formation of hydrogen bonds between amide or carbamide carbonyl functions and nitric acid.

After contact extraction reaction with uranyl ion, C=O_uranyl_ stretching band arose at 1497 cm^−1^ and 1502 cm^−1^ after U(VI) extraction for ***L1*** and ***L2***, respectively. In addition to this, another new peak that appeared at 936 cm^−1^ is a uranyl stretching frequency [31,32]. It is generally believed that the separation (Δν) of the asymmetric and symmetric ONO stretching frequencies can be used to determine the mode of nitrate coordination with the metal. A Δν larger than 186 cm^−1^ indicates a bidentate chelate environment, while a Δν of 115 cm^−1^ or lower indicates monodentate coordination [33]. The bands at around 1530 and 1282−1283 cm^−1^ are characteristic of nitrate asymmetric and symmetric stretching bands. The difference between these two bands larger than 186 cm^−1^ suggests bidentate coordination of nitrate toward uranium. 

Upon coordination to uranium, the C=O vibration band is expected to shift to lower wavenumbers. For tetra-alkylcarbamides, some studies have clarified that the peak attribution for the carbonyl stretching frequency is not straightforward [19]. The shift in carbonyl frequency upon uranyl binding is greater than that for monoamide, which will lead to a broader peak appearing in the range of 1495–1576 cm^−1^. In Figure 6, some peaks appear in the range of 1500–1560 cm^−1^, which can indicate that C=O is involved in coordination. The above results show that the predominant uranyl species formed upon uranyl extraction with ***L*** is bidentate coordination of nitrate toward uranium and monodentate coordination with ***L*** and uranium which is consistent with the findings of other tetra-alkylcarbamides ligands.

### 3.3. Thermodynamic Calculation

To obtain the thermodynamic data in the extraction process, the extraction of U(VI) with two ligands was studied at the temperature range of 303–340.5 K (Figure 7). According to the Van’t Hoff equation,
(3)ΔG=ΔH– TΔS=–2.303RTlogK
(4)$$logK=–\frac{\Delta H}{2.303RT}+\frac{\Delta S}{2.303R}$$
where, *R* is the gas constant, and *T* denotes absolute temperature. Δ*G* is Gibbs energy change and Δ*S* is entropy change. Approximating *D_U_* as *K*, a plot of log*D_U_* vs. 1/*T* gives a straight line with a slope of −Δ*H/2.303R*, and an intercept of Δ*S/2.303R*. The detailed information of thermodynamic data is shown in Table 1. The negative values of Δ*H* for all ligands show that the complexations of U(VI) by ligands molecules are exothermic, whereas the values of Δ*G* at ambient temperature indicate that the extraction events are spontaneous.

### 3.4. DFT Studies

#### 3.4.1. Geometry Analysis

The structure of the complexes was simulated. All the optimized coordination structures were proven to be local minima by harmonic vibrational frequency calculations. Figure 8 shows the minimum energy structures of UO_2_(NO_3_)_2_·2***L*** complexes in the solution phase. The coordination number of uranium ions is 6 with the nitrate anions in a bidentate mode.

Table 2 shows the computed structural characteristics for the U(VI) complexes in both gas and solution phases. There are few differences between bond lengths in these two phases. In general, the U−O(L) bonds are usually shorter than the U−O(NO_3_^−^) bonds, and U(VI) in both complexes bind more tightly to ligand than NO_3_^−^. Furthermore, the U−O(L) bonds in U−***L1*** complexes are shorter than those in U−***L2*** complexes, indicating that ***L1*** has a better capacity to group with U(VI) ions. These are consistent with the experimental results that ***L1*** has a stronger uranium extraction impact than ***L2***.

#### 3.4.2. NBO Analysis

Wiberg Bond Indices (WBIs), which have been recognized as useful criteria to evaluate the degree of covalency in previous reports, [34,35,36] were used to examine the bond order of U−O(***L***) and U−O(NO_3_^−^) bonds. The WBIs of U−O(L) in the solution phase are greater than those in the gas phase, as shown in Table 3 the case of U−O(NO_3_^−^) is in opposition. The WBIs are about 0.70 for U−O(***L***) and 0.59 for U−O(NO_3_^−^) in the gas phase and are 0.72 for U−O(***L***) and 0.60 for U−O(NO_3_^−^) in the solution phase. Such WBIs values, especially for the U−O(NO_3_^−^) bond, indicate the electrostatic interactions between metal ions with ligand and NO_3_^−^.

Table 4 shows the natural charges on uranium and oxygen atoms (both oxygen of ligands and NO_3_^−^) in UO_2_(NO_3_)_2_·2***L***. The natural charges on the U atoms are much smaller than those of the UO_2_^2+^ suggesting important ligand-to-metal charge transfer in these species. Besides, the O(***L***) atom has a more negative charge than the O(NO_3_^−^) atom, suggesting the strongest electron-donating abilities.

#### 3.4.3. Topological Analysis

The QTAIM approach was used to investigate the bond properties behavior of ***L*** with U(VI) [26,37,38].The electron density (ρ) and the Laplacian of the electron density (∇^2^ρ) at the bond critical points (BCPs) may provide valuable information about the strength and characteristics of the bonds [39,40].

Generally, the values of ρ at the BCP > 0.20 a.u. and ∇^2^ρ < 0 refer to a typical covalent bond. In contrast, ρ < 0.10 a.u. and ∇^2^ρ > 0 a.u. reflect an ionic bond [41]. The data for ρ and ∇^2^ρ at the BCPs of U−***L*** complexes are listed in Table 5.

In this study, the values of 0 < ρ < 0.1, and ∇^2^ρ > 0 refer to an ionic bond [42]. The values of ρ at the U−O(***L***) BCPs of both complexes are slightly bigger than those at the U−O(NO_3_^−^), implying that the U−O(***L***) bonds have a stronger capacity for complexation. The calculated delocalization index (δ), which is considered a bond order index, for U−O(***L***) bond critical points in UO_2_(NO_3_)_2_·2***L*** (nearly 1.1) is relatively larger compared to that of U−O(NO_3_^−^) (nearly 0.9). Both kinds of bonds are single bonds, which agree with the bond order analysis.

## 4. Conclusions

A study of uranium(VI) extraction by the two novel tetra-alkylcarbamides was performed. As the nitric acid concentration increased from 0.1 to 8 mol/L, the *D_U_* increased firstly and then decreased. The maximum *D_U_* appeared at 5 mol/L nitric acid concentration with 5.02 and 4.93 for ***L1*** and ***L2***, respectively with no third phase formation. This result was competitive with monoamides. Additionally, thermodynamic studies showed that the extraction process was a spontaneous exothermic reaction. The structure of complexes was determined to be UO_2_(NO_3_)_2_·2***L*** via slope analysis. Furthermore, FT-IR results showed that the complexes formed by bidentate coordination of nitrate toward uranium and monodentate coordination with ***L*** and uranium. DFT calculations have also been applied to build the structures of UO_2_(NO_3_)_2_·2***L*** complexes, bonding analysis indicated that ***L*** and U(VI) were predominantly through ionic interactions. The short carbon branched ***L1*** had a better capacity to group with U(VI) than ***L2***. This work enriches the study of the coordination behavior of tetra-alkylcarbamides with uranium, and also provides guidance for the designation of the uranium extraction ligand in the future.

## Figures and Tables

**Figure 1 molecules-27-05527-f001:**
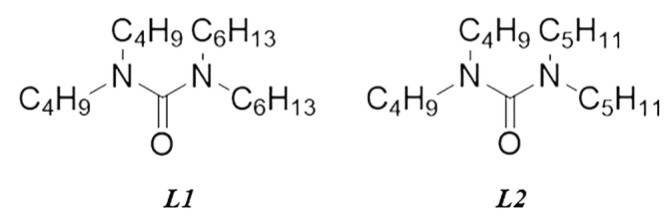
Structures of two novel tetra-alkylcarbamides.

**Figure 2 molecules-27-05527-f002:**
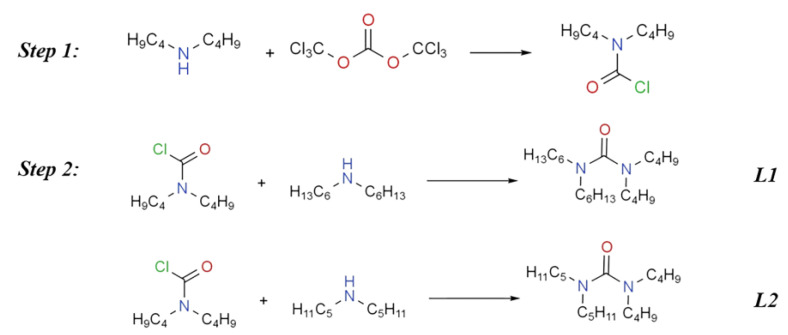
Reaction scheme for the synthesis of ***L***.

**Figure 3 molecules-27-05527-f003:**
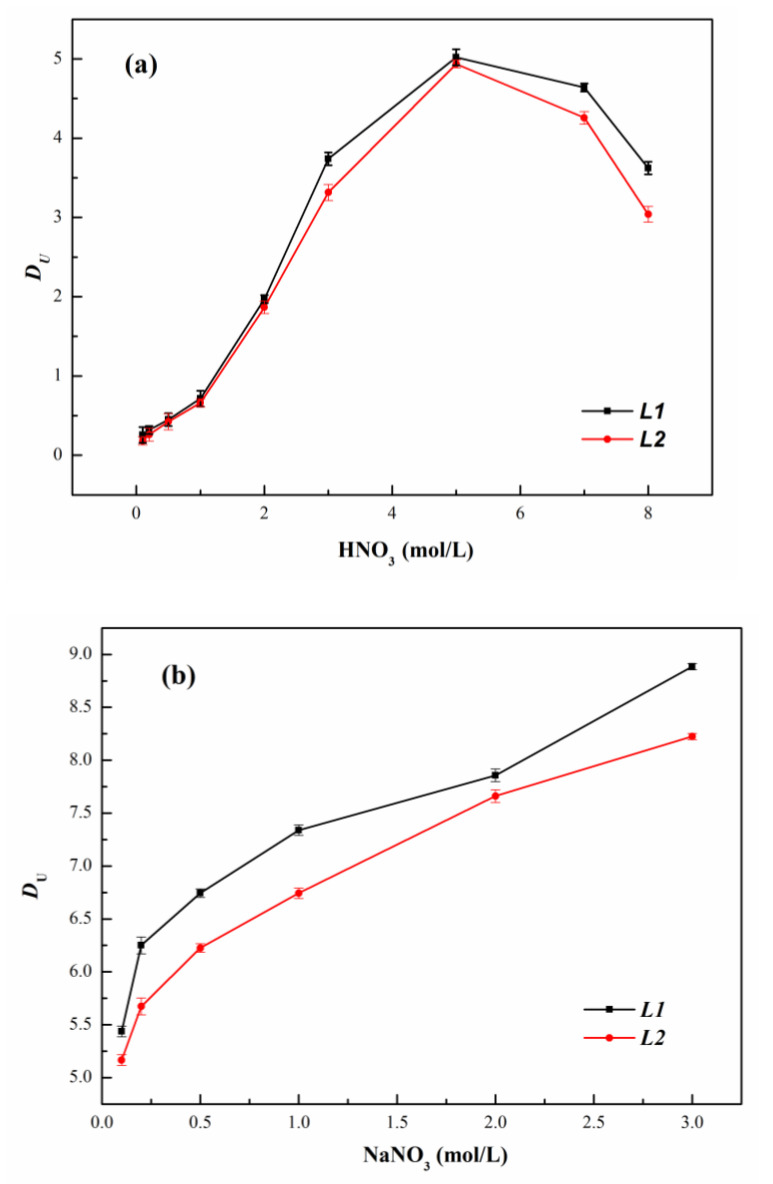
*D_U_* by ***L*** in n-dodecane as a function of (**a**) the nitric acid concentration ([U(VI)]_ini_ = 10 g/L) (**b**) the NaNO_3_ concentration ([HNO_3_]_ini_ = 5 mol/L, [U(VI)]_ini_ = 10 g/L).

**Figure 4 molecules-27-05527-f004:**
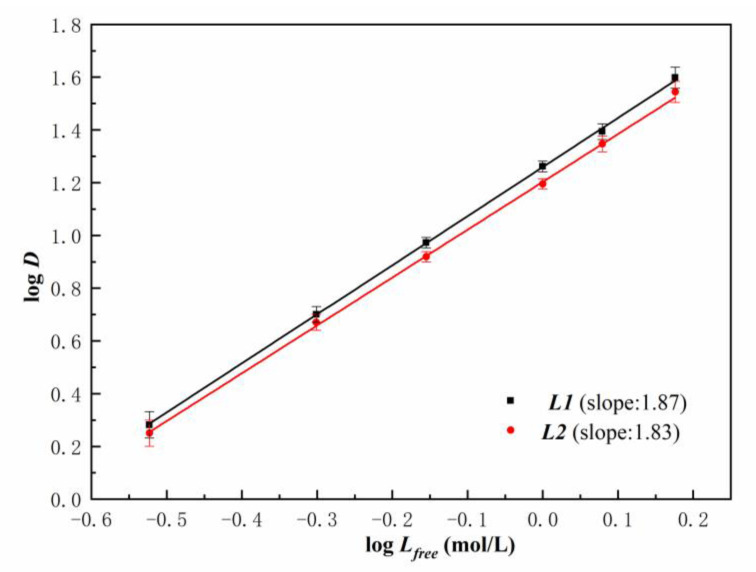
D*_U_* by two ligands in n-dodecane as a function of ***L*** concentration ([HNO_3_]_ini_ = 5 mol/L, [U(VI)]_ini_ = 10 g*/*L).

**Figure 5 molecules-27-05527-f005:**
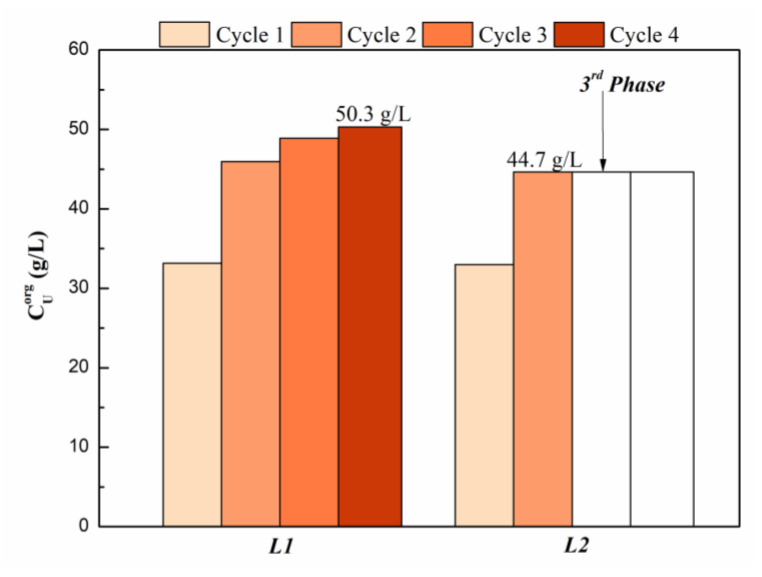
Uranium(VI) loading capacity of ***L***.

**Figure 6 molecules-27-05527-f006:**
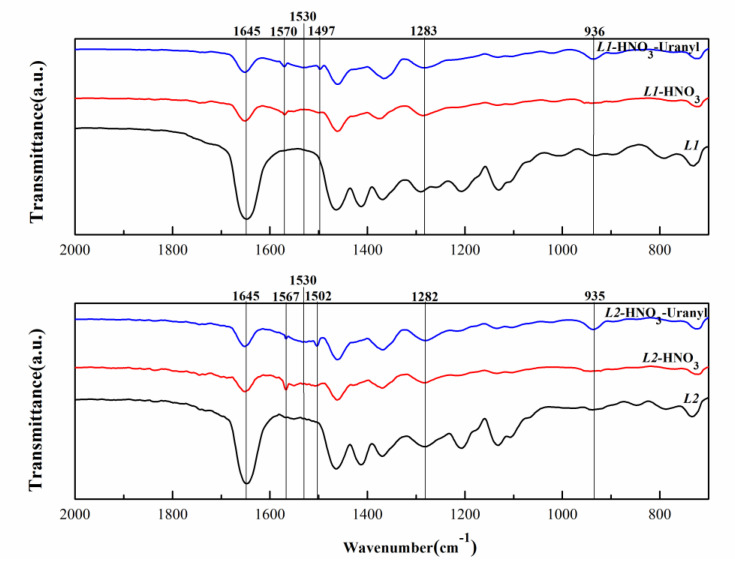
Infrared spectra of organic phases: solutions of ***L***, solutions contacted with nitric acid ([HNO_3_]_ini_ = 3 mol/L), and U(VI) aqueous solutions with nitric acid ([HNO_3_]_ini_ = 3 mol/L).

**Figure 7 molecules-27-05527-f007:**
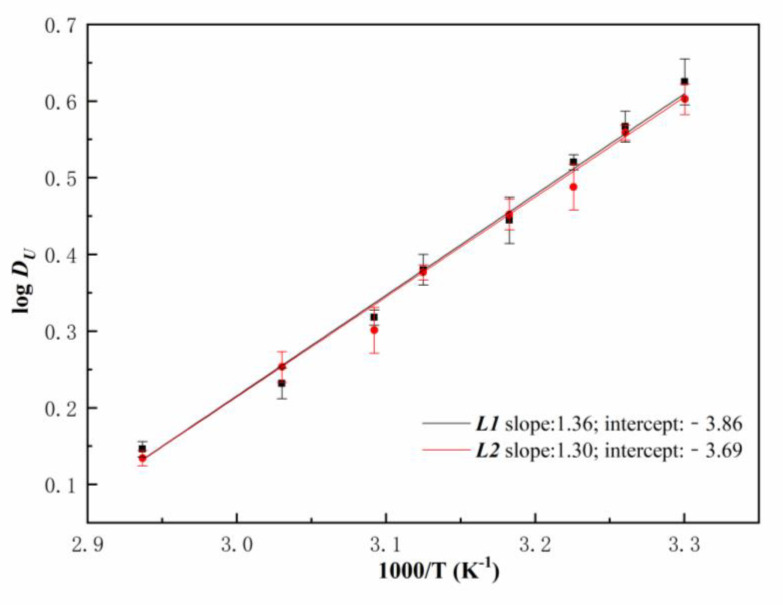
D*_U_* by two ligands in n-dodecane as a function of temperature ([HNO_3_]_ini_ = 5 mol/L, [U(VI)]_ini_ = 10 g*/*L).

**Figure 8 molecules-27-05527-f008:**
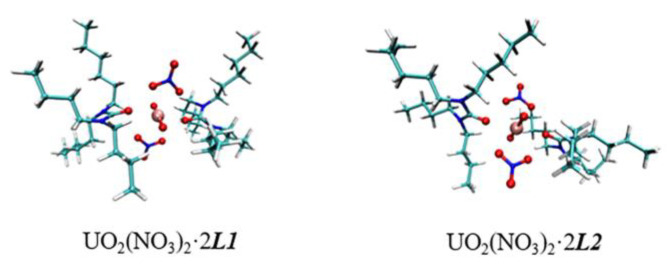
Optimized geometries of free ***L*** and U-***L*** complexes.

**Table 1 molecules-27-05527-t001:** Thermodynamic data of ***L***/n-dodecane extraction systems.

Thermodynamic Data	*L*1	*L*2
Δ*H* (kJ/mol)	−25.97	−24.89
Δ*S* (J/mol K)	73.91	70.70
Δ*G* (kJ/mol)	−47.99	−45.96

**Table 2 molecules-27-05527-t002:** Average bond lengths(Å) of two complexes after extraction in both gas phase and solution phase.

Complexes	Gas Phase		Solution Phase (hexane)	
	U−O(*L*)	U−O(NO_3_^−^)	U−O(*L*)	U−O(NO_3_^−^)
U−*L1*	2.407	2.540	2.395	2.547
U−*L2*	2.413	2.536	2.397	2.545

**Table 3 molecules-27-05527-t003:** Wiberg Bond Indices (WBIs) at the B3LYP/ECP60MWB/6-311g(d,p) level of theory in both gas phase and solution phase.

Complexes	Gas Phase		Solution Phase (hexane)	
	O(*L*)	O(NO_3_^−^)	O(*L*)	O(NO_3_^−^)
U−*L1*	0.702	0.593	0.725	0.585
U−*L2*	0.689	0.599	0.717	0.586

**Table 4 molecules-27-05527-t004:** NPA charges on the Uranium and Oxygen atoms in UO_2_(NO_3_)_2_·2***L*** species.

Complexes	U	*L*		NO_3_^−^			
		O1	O2	O3	O4	O5	O6
UO_2_(NO_3_)_2_·2*L1*	1.347	−0.644	−0.645	−0.461	−0.437	−0.462	−0.437
UO_2_(NO_3_)_2_·2*L2*	1.290	−0.625	−0.629	−0.427	−0.451	−0.459	−0.435

**Table 5 molecules-27-05527-t005:** QTAIM analyses of ρ (e^−^/Bohr), ∇^2^*ρ* (e^−^/Bohr), and δ at BCPs of two complexes at the B3LYP/ECP60MWB/6-311g(d,p) level of theory in solution phase.

Parameters	U−*L1*		U−*L2*	
	U−O(*L*)	U−O(NO_3_^−^)	U−O(*L*)	U−O(NO_3_^−^)
ρ	0.058	0.048	0.056	0.048
∇^2^*ρ*	0.254	0.165	0.257	0.165
δ	1.131	0.935	1.120	0.943

## Data Availability

Not applicable.

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
