# Peer review of "Extraction of Uranium in Nitric Media with Novel Asymmetric Tetra-Alkylcarbamide"

_molecules, 2022, doi:10.3390/molecules27175527_

Round 1

Reviewer 1 Report

This article concerns a study of the uranium extraction of two novel asymmetric tetra-alkyl carbamide extractants that differ from their alkyl chain length.

This manuscript encompasses a large body of experimental and theoretical work, with extraction experiments as a function of time, nitric acid and ligand concentration, FTIR analysis and thermodynamic studies to characterize the structural organization of the system . A very interesting  theoritical study allows confirming and interpreting the results.

This works is worth being published in Molecules, however minor points have to be completed before its acceptance.

Page 1

Title should be completed and mention that extraction of uranium only concern nitroc media, and that the tested tetra-alkyl carbamide extractants are asymmetric.

It would be interesting to mention the targetted application of such ligands in the abstract (to enhance uranium separation, enrichment and recovery in the nuclear fuel cycle)

Please explain in more what is meant in the following sentence : « also exist wherein U(VI) is co-extracted along with the other actinides. »

Page 2

« Two ligands show a very high U(VI) loading capacity without any third phase formation even at high HNO3 concentration, and their viscosity remains acceptable for process development. As a result, their potential for use in uranium (VI) extraction in the post-treatment process is demonstrated. »

For sake of homogeneity, this sentence should be written in the past tense.

It would be interesting to mention which are the two interesting ligands of the literature, and to compare them to the one of the study.

« Theoretically, a larger distribution ratio may be achieved by designing four different substituent structures. »

Please explain and jusity this asumption.

Figure 1 : the shorter Ligand should be presented in first. And the two ligands should be illustrated similarly : with their shorter chains on the left side.

Page 4

« With longer alkyl chains, as illustrated in Figure 3a, although slight, the DU becomes bigger and follows the sequence L1>L2. For both monoamides[27-29] and tetra-alkylcarbamides,[18] the effect of alkyl chain length has been studied earlier. A longer carbon branch chain can improve the solubility of the complex in the organic phase, in order to boost the DU. »

Please specify that the longer ligand is L1.

It would be usefull to give details on the ligands structure that were investigated in ref 27, 29 and 18. Could a comparison between lignad L1 and L2 with the ones of literature provide information on effect of alkyl chain length ? on effect of alkyl chain asymmetry ?

Page 5

Figure 3 : a comparison with a symmetric Ligand would have been interesting to show, or at least to discuss in the text. Is the molecule of ref 18 symmetric with similar alkyl chain length ?

Page 6

Figure 4 : Please show the fitted line instead of artificial connexion between the experimental points.

Please provide more details on the method employed to derive the complexe stoichiometry. For instance, Slope analysis should be performed as a function of free ligands as in doi.org/10.1080/07366299.2020.1790527

Please show the Du as a function of Ligand concentration in Supplementary material (with no log).

The loading capacity would have been interesting to evaluate and to present in this paper. It is a very convincing parameter to compare extractant efficiency. It could clearly allow providing a clear comparision between the covnetional extractant TBP, carbamide from literature and the two ligand L1 and L2. Is it possible to add ?

Page 8

As for Figure 4, please show the fitted lines.

Page 10

« This work enriches the study of the coordination behavior of tetra-alkylcarbamides with uranium, and also provides guidance for the designation of the uranium extraction ligand in the future. »

Could this work allow predicting the effect of asymmetry for a future extractant design ? For higher extraction efficiency, would the authors recommend longer, shorter or asymmetric chains ? In which extent ? Such discussions would increase the interest of the study, and broaden the readership of this valuable work.

Author Response

Journal: Molecules

Manuscript ID: molecules-1847422

Title: Extraction of uranium with novel tetra-alkylcarbamide

Authors: Qi Chen, Baole Li, Junli Wang, Haowei Zhu, Xiwen Chen, Yifu Hu, Jia Zhou, Xiang Li, Weifang Zheng, Taihong Yan*

Dear editor and Reviewer:

Thank you for your letter and for the comments concerning our manuscript. Those comments are all valuable and very helpful for revising and improving our paper, as well as the important guiding significance to our researches. We have studies comments carefully and have made correction which we hope meet with approval. Revised portion are marked in red in the paper. The main correction in the paper and the responds to the reviewer’s comments are as flowing:

Q1. Title should be completed and mention that extraction of uranium only concern nitric media, and that the tested tetra-alkyl carbamide extractants are asymmetric.

Response: Thank you for the reviewer’s valuable comments. The title was changed to: Extraction of uranium in nitric media with novel asymmetric tetra-alkylcarbamide.

Q2. It would be interesting to mention the targetted application of such ligands in the abstract (to enhance uranium separation, enrichment and recovery in the nuclear fuel cycle).

Response: We thank the reviewer’s helpful comments. We have modified the abstract section accordingly: This study enriches the coordination chemistry of U(VI) by tetra-alkylcarbamides, which may offer new clues for the design and synthesis of novel ligands for the separation, enrichment and recovery of uranium in the nuclear fuel cycle.

Q3. Please explain in more what is meant in the following sentence : « also exist wherein U(VI) is co-extracted along with the other actinides.

Response: This phrase appears at the end of the second paragraph of the introduction section. The first sentence of the paragraph points out several extractants that have good extraction properties for U(VI). However, they tend to be poorly selective, and some reports show that they have good extraction ability for all other actinides. Take CMPO as an example, this class of ligand also has good extraction for Th, Am and Pu. (Ref: J.N. Mathur , M.S. Murali , P.R. Natarajan, L.P. Badheka, A. Banerji. Extraction of actinides and fission products by octyl(phenyl)-N,N-diisobutylcarbamoylmethyl-phosphine oxide from nitric acid media.[J]. Talanta, 1992, 39(5):493-496. DOI: 10.1016/0039-9140(92)80170-I)

To make the meaning clearer, replace this sentence with: However, P-containing ligands would seriously pollute the ecosystem, and the selective extraction capacity of U(VI) also needs to be improved.

Q4. Two ligands show a very high U(VI) loading capacity without any third phase formation even at high HNO3 concentration, and their viscosity remains acceptable for process development. As a result, their potential for use in uranium (VI) extraction in the post-treatment process is demonstrated.

For sake of homogeneity, this sentence should be written in the past tense.

Response: We thank the reviewer for the careful suggestions. The new sentence has been changed as: Two ligands showed a very high U(VI) loading capacity without any third phase formation even at high HNO3 concentration, and their viscosity remained acceptable for process development. As a result, their potential for use in uranium (VI) extraction in the post-treatment process was demonstrated.

Q5. « Theoretically, a larger distribution ratio may be achieved by designing four different substituent structures. » Please explain and jusity this asumption.

Response: Thank you for the reviewer’s valuable comments. By re-finding the references and understanding it, it is now believed that: Theoretically, asymmetric ligand structure with both long-chain alkyl groups and short-chain alkyl groups has a more diverse structure, consequently reducing the spatial resistance and can thus obtain a higher distribution ratio while improving the solubility of the complexes in diluents and alleviating or avoiding the formation of third-phase during extraction. Asymmetric DGAs have been well developed in this regard, like DMDODGA etc. Ref: (J. Ravi, T. Prathibha, K.A. Venkatesan, M.P. Antony, T.G. Srinivasan, P.R. VasudevaRao, Third phase formation of neodymium (III) and nitric acid in unsymmetrical N, N-di-2-ethylhexyl-N', N'-dioctyldiglycolamide, Sep. Purif. Technol. 85 (2012) 96–100, DOI: 10.1016/j.seppur.2011.09.053.) (Sasaki Y, Sugo Y, Suzuki S, Tachimori S (2001) The novel extractants diglycolamides, for the extraction of lanthanides and actinides in HNO3-n-dodecane system. Solvent Extr Ion Exch 19:91–103. DOI: 10.1081/SEI-100001376.)

For greater clarity, this sentence reads: Theoretically, the design of four different substituent structures (combination of long and short carbon chains) can increase the solubility of the complex in the diluent while maintaining a high distribution ratio thereby reducing or preventing the generation of a third phase during extraction.

Q6. Figure 1: the shorter Ligand should be presented in first. And the two ligands should be illustrated similarly : with their shorter chains on the left side.

Response: We thank the reviewer for the careful suggestions. Figure 1 has been modified accordingly.

Q7. « With longer alkyl chains, as illustrated in Figure 3a, although slight, the DU becomes bigger and follows the sequence L1>L2. For both monoamides[27-29] and tetra-alkylcarbamides,[18] the effect of alkyl chain length has been studied earlier. A longer carbon branch chain can improve the solubility of the complex in the organic phase, in order to boost the DU. »

Please specify that the longer ligand is L1.

Response: We thank the reviewer for the careful suggestions. To make it clearer, the sentence was changed to: For L1 with a longer alkyl chain, as illustrated in Figure 3a, although slight, the DU becomes bigger and follows the sequence L1>L2.

Q8. It would be usefull to give details on the ligands structure that were investigated in ref 27, 29 and 18. Could a comparison between ligand L1 and L2 with the ones of literature provide information on effect of alkyl chain length ? on effect of alkyl chain asymmetry?

Response: Thank you for the reviewer’s valuable comments. In the last paragraph on page 4 we have added some details on the ligands structure information of the reference ligand. < For example, for the extraction study of N-methyl, N-butyl derivatives of hexanamide (MBHA), octanamide (MBOA) and decanamide (MBDA), the extraction of the U(VI) follows the order of MBHA < MBOA < MBDA.>

A comparison of the extraction results with Ref 18 (symmetrical ligands) is shown in the last paragraph on page 5. <Compared with similar carbon chain length symmetric ligand N,N,N’,N’-tetra-n-butylurea (TBU), 1.2 mol/L TBU in the TPH diluent system with a nitric acid concentration of 0.5 mol/L had DU = 0.6. The results of this study are very competitive, especially since they were achieved at a ligand concentration of only 0.5 mol/L>. Since the experimental conditions are not identical, it is not possible to obtain a very visual comparison in talking about the effect of the asymmetry of the ligand on the partition ratio.

Q9. Figure 3 : a comparison with a symmetric Ligand would have been interesting to show, or at least to discuss in the text. Is the molecule of ref 18 symmetric with similar alkyl chain length ?

Response: Thank you for the reviewer’s valuable comments. Four symmetrical ligands with different carbon chain lengths are discussed in Ref 18 (TBU, TPU, THU, TOU):

Unfortunately, the DU values under multiple acidity conditions were not reported in ref 18. Only two acidity conditions, 0.5 mol/L and 4 mol/L, were chosen in the article for the calculation of DU values. The low acidity(0.5 mol/L) that is more favorable for the current study was selected for comparison and additional analysis: Compared with similar carbon chain length symmetric ligand N,N,N’,N’-tetra-n-butylurea (TBU), 1.2 mol/L TBU in the TPH diluent system with a nitric acid concentration of 0.5 mol/L had DU = 0.6. The results of this study are very competitive, especially since they were achieved at a ligand concentration of only 0.5 mol/L.

Q10. Figure 4 : Please show the fitted line instead of artificial connexion between the experimental points.

Please provide more details on the method employed to derive the complexe stoichiometry. For instance, Slope analysis should be performed as a function of free ligands as in doi.org/10.1080/07366299.2020.1790527

Response: Figure 4 has been modified accordingly. We thank the reviewers for recommending the article, and we have used it as a reference to add a more detailed computational derivation to section 2.3 in the Supplemental Information. Ref 30 was also replaced as well. (New Ref 30: A. Artese, S. Dourdain, N. Felines, G. Arrachart, N. Boubals, P. Guilbaud, S. Pellet-Rostaing, Bifunctional Amidophosphonate Molecules for Uranium Extraction in Nitrate Acidic Media. Solvent extraction and ion exchange. 2020, DOI: 10.1080/07366299.2020.1790527.)

Q11. Please show the Du as a function of Ligand concentration in Supplementary material (with no log).

Response: Thank you for the reviewer’s valuable comments. The Figure S7 has been added to the section 2.2 in the Supplement Information.

Q12. The loading capacity would have been interesting to evaluate and to present in this paper. It is a very convincing parameter to compare extractant efficiency. It could clearly allow providing a clear comparision between the covnetional extractant TBP, carbamide from literature and the two ligand L1 and L2. Is it possible to add ?

Response: Thank you for the reviewer’s valuable comments. We supplemented the loading experiments with two ligands. An experimental discussion of the load capacity has been also added to section 3.1.3 : The research of the maximum uranium loading capacity in organic phases was carried out by 0.5 mol/L L in organic solutions which were contacted four times with fresh uranium solution of U(VI) 43.9 g/L (0.18 mol/L) in 5 mol/L nitric acid. The uranium loading concentration after each cycle is reported in Figure 5.

Figure 5. Uranium(VI) loading capacity of L. (Figure 5 is in the revised manuscript and cannot be uploaded here)

For the L1 ligand with longer carbon chains, no third-phase occurred in all four cycles, and the loading of U in the organic phase reached 50.3 g/L after four cycles. For the L2 ligand with a shorter carbon chain, third-phase appeared after the third cycle, and the loading of U in the organic phase after the second cycle was 44.3 g/L. Compared to the reported results, 1.2 mol/L TBU showed a third-phase in the first cycle of contact with 200 g/L uranium[18]. The asymmetry of the ligands seems to have mitigated the third-phase generation.

As much as 0.21 mol/L (50.3 g/L) of uranium can be extracted in organic phases, which corresponds to a CL1/CU(VI) ratio of 2.4. This result indicates that the L1 should form a 1:2 uranyl/carbamide complex as major species in the organic phase, which is consistent with the results obtained by slope analysis.

Q13. Page 8. As for Figure 4, please show the fitted lines.

Response: We thank the reviewer for the careful suggestions. Figure 7 has been modified accordingly.

Q14. This work enriches the study of the coordination behavior of tetra-alkylcarbamides with uranium, and also provides guidance for the designation of the uranium extraction ligand in the future. »

Could this work allow predicting the effect of asymmetry for a future extractant design ? For higher extraction efficiency, would the authors recommend longer, shorter or asymmetric chains ? In which extent ? Such discussions would increase the interest of the study, and broaden the readership of this valuable work.

Response: In this study, it was found that ligands with longer carbon chains have slightly higher distribution ratios. And by comparing the loading experiments of symmetric ligands, it can be seen that the asymmetry structure might resist the generation of third phases. In order to better discuss the advantages of asymmetric ligand structures. Follow-up experiments will consider ligand structures with longer carbon chains, such as 1,1-dibutyl-3,3-dioctylurea; Or ligands that possess four more disparate carbon chain structures: 1,1-diethyl-3,3-dioctylurea, etc.

Reviewer 2 Report

This manuscript was well written an appropriately outline the U(IV) extraction behaviour of two tetra-alkylcarbamide ligands using solvent extraction methods.  A few minor comments and clarifications are asked for.

1) Introduction first paragraph- the term kilos is used. Please change to kilograms.

2) Introduction second paragraph- I'm not sure what is mean by "...and also exist wherein U(VI) is co-extracted along with other actindes."  Please clarify the language

3) Introduction fourth paragraph- Don't need to start with A.  Just say "Research has been conducted with..."

4) Experimental section 2.1- Describe grade of acids and chemicals.  Seem to be missing a few chemicals used in the synthesis steps- please add.

5) Section 2.3- Perhaps a table descirbing each extraction experiment would be helpful.  It is unclear when the variable changes what was actually done.  For example in effects of [HNO3]- there is nothing stating what [HNO3] were were used.  Based on the figure I guess it was 8,7,5,3,2,and lower- please state actual concentrations.  Same comment for NaNO3 concentrations.

6) I assume the monamide studies referenced also used a 10g/L U solution.  PLease verify for consistency of the experiment or explain difference.

7) the slope of the LogD vs LogL is 1.87 and 1.83 for each ligand.  The results then say that the stoiciometric ratio is 2.  Is that based on uncertainty analysis or is it common to round up?  I've seen where partial fractions can be reported for ratio so am curious as to why 2 is described and not 1.8.

8) Equation 4 list lgK- please verify this is appropriate format for journal.  The text below states, "A plot of lgDu vs 1/T gives..." however the plot says LogD.  PLease be consistent if you mean lgDu or LogD.

9) The conclusions sate that the Du is 5.24 and 4.85, please add uncertanties and/or confidences to those values.  In the discussion it says one is higher- was that statistically determined?

Author Response

Journal: Molecules

Manuscript ID: molecules-1847422

Title: Extraction of uranium with novel tetra-alkylcarbamide

Authors: Qi Chen, Baole Li, Junli Wang, Haowei Zhu, Xiwen Chen, Yifu Hu, Jia Zhou, Xiang Li, Weifang Zheng, Taihong Yan*

Dear editor and Reviewer:

Thank you for your letter and for the comments concerning our manuscript. Those comments are all valuable and very helpful for revising and improving our paper, as well as the important guiding significance to our researches. We have studies comments carefully and have made correction which we hope meet with approval. Revised portion are marked in red in the paper. The main correction in the paper and the responds to the reviewer’s comments are as flowing:

Q1. Introduction first paragraph- the term kilos is used. Please change to kilograms.

Response: We thank the reviewer for the careful suggestions. The corresponding changes have been made.

Q2. Introduction second paragraph- I'm not sure what is mean by "...and also exist wherein U(VI) is co-extracted along with other actindes."  Please clarify the language.

Response: Thank you for the reviewer’s valuable comments. To make the meaning clearer, this sentence has been changed to: However, P-containing ligands would seriously pollute the ecosystem, and the selective extraction capacity of U(VI) also needs to be improved.

Q3. Introduction fourth paragraph- Don't need to start with A.  Just say "Research has been conducted with..."

Response: We thank the reviewer for the careful suggestions. The corresponding changes have been made.

Q4. Experimental section 2.1- Describe grade of acids and chemicals.  Seem to be missing a few chemicals used in the synthesis steps- please add.

Response: We thank the reviewer for the careful suggestions. Additions have been made in section 2.1 : Aladdin provided the nitric acid as well as all the reagents used in the ligand synthesis section.

Q5. Section 2.3- Perhaps a table descirbing each extraction experiment would be helpful.  It is unclear when the variable changes what was actually done. For example in effects of [HNO3]- there is nothing stating what [HNO3] were used. Based on the figure I guess it was 8,7,5,3,2, and lower- please state actual concentrations. Same comment for NaNO3 concentrations.

Response: We thank the reviewer’s helpful comment. In order to describe more clearly the experimental conditions of nitric acid as well as sodium nitrate in the experiment, this part is added in section 2.3: The dependence of HNO3 concentration: 0.1 mol/L, 0.2 mol/L, 0.5 mol/L, 1 mol/L, 2 mol/L, 3 mol/L, 5 mol/L, 7 mol/L, 8 mol/L. The dependence of NaNO3 concentration: 0.1 mol/L, 0.2 mol/L, 0.5 mol/L, 1 mol/L, 2 mol/L, 3 mol/L.

Q6. I assume the monamide studies referenced also used a 10g/L U solution.  PLease verify for consistency of the experiment or explain difference.

Response: We thank the reviewer for the careful suggestions. The experimental data on monoamide in this article were obtained from Ref 18 (C. Berger, C. Marie, D. Guaillaumont, E. Zekri, L. Berthon, Extraction of uranium(VI) and plutonium(IV) with tetra-alkylcarbamides. Solvent extraction and ion exchange. 2019, 37(2), 111–125. DOI: 10.1080/07366299.2019.1630095). In which the U concentration in aqueous phase was 10 g/L according to the description (the illustration in Figure 3 of Ref 18).

Q7. The slope of the LogD vs LogL is 1.87 and 1.83 for each ligand.  The results then say that the stoiciometric ratio is 2.  Is that based on uncertainty analysis or is it common to round up?  I've seen where partial fractions can be reported for ratio so am curious as to why 2 is described and not 1.8.

Response: We thank the reviewer for the careful suggestions. The conclusion that the stoichiometric number is 2 was obtained by rounding. For clarity of presentation, the sentence has been changed to: Indicating that the stoichiometric ratio of U(VI) to L in the extracted species is mainly in the form of 1:2, as the major complexation formed as UO2(NO3)2·2L.

Q8. Equation 4 list lgK- please verify this is appropriate format for journal.  The text below states, "A plot of lgDu vs 1/T gives..." however the plot says LogD.  PLease be consistent if you mean lgDu or LogD.

Response: We thank the reviewer for the careful suggestions. lgK in the Equation 4 has been changed into logK. The latter expression < lgDu >changed to logDU. The vertical coordinate of Figure 6 becomes logDU.

Q9. The conclusions sate that the Du is 5.24 and 4.85, please add uncertanties and/or confidences to those values.  In the discussion it says one is higher- was that statistically determined?

Response: We thank the reviewer for the careful suggestions, after comparing the data results in section 3.1.1, the conclusion section has now been changed to: The maximum DU appears at 5 mol/L nitric acid concentration with 5.02 and 4.94 for L1 and L2, respectively with no third phase formation.

As written in section 2.3: Two aliquots of the aqueous raffinate were collected for concentration determination during double extraction procedures under the same experimental conditions. As a result, each distribution ratio was calculated by averaging several samples, with an error of less than 5%.” Also in section 2.4: Each sample was repeated two times, with the average utilized for further study.

The experimental data are not only from multiple sets of parallel experiments, but the measurements are also multiple times. The comparison between the data is done by comparing between the average values. By calculating the uncertainty by the initial data, the data obtained as: 5.02±0.24 and 4.93±0.24.